# Rates of Mitochondrial Metabolism of Glucose, Amino Acids, and Fatty Acids by the HEI-OC1 Inner Ear Cell Line

**DOI:** 10.3390/biology14091118

**Published:** 2025-08-24

**Authors:** Kento Koda, Teru Kamogashira, Ken Hayashi, Chisato Fujimoto, Shinichi Iwasaki, Tatsuya Yamasoba, Kenji Kondo

**Affiliations:** 1Department of Otolaryngology and Head and Neck Surgery, Faculty of Medicine, University of Tokyo, 7-3-1, Hongo, Bunkyo-ku, Tokyo 113-8655, Japan; 2Ikebukuro ENT Hayashi Clinic, 2F, Toshima Eco-Musee Town, 2-45-3, Minami-Ikebukuro, Toshima-ku, Tokyo 171-0022, Japan; 3Department of Otolaryngology and Head and Neck Surgery, Faculty of Medicine, Nagoya City University, Kawasumi 1, Mizuho-cho, Mizuho-ku, Nagoya City 467-8601, Aichi, Japan; 4Tokyo Teishin Hospital, 2-14-23 Fujimi, Chiyoda-ku, Tokyo 102-0071, Japan

**Keywords:** oxygen consumption rate, HEI-OC1, mitochondrial metabolic rate, glucose, age-related hearing loss

## Abstract

Mitochondria, the cell’s energy-producing units that utilize fuels such as glucose, amino acids, and fatty acids, play a crucial role in cellular metabolism. Aging is known to impair mitochondrial function, leading to decreased energy production and increased cellular vulnerability. Cochlear cells—the sensory cells responsible for hearing—are particularly susceptible to age-related metabolic decline, which contributes to hearing loss. However, the specific changes in energy metabolism within cochlear cells during aging remain unclear. The aim of this study was to investigate how different metabolic fuels influence mitochondrial energy production in cochlear cells, focusing on their capacity to support cellular respiration under varying energy demands. Oxygen consumption rates were measured to assess mitochondrial activity following the addition of various substrates. The results showed that glutamine (L-Gln) and pyruvate significantly boosted energy production more than other fuels, such as glucose and other amino acids, while palmitate increased oxygen consumption without substantially contributing to ATP production. L-Gln, in particular, was found to be more effective than other amino acids in supporting energy needs. In summary, cochlear cells primarily rely on glucose for basal energy metabolism, but pyruvate and L-Gln serve as key substrates during acute energy demands, whereas fatty acids have a limited role. Understanding these substrate-dependent metabolic responses may inform future strategies for maintaining cochlear function.

## 1. Introduction

Age-related hearing loss (ARHL), known as presbycusis, is one of the most common sensory organ diseases. It affects one-third of individuals over 65 years of age in modern society, which is referred to as a super-aging society, and is closely linked to the development of cognitive decline, social isolation, and depression [1,2,3,4]. Unmanaged hearing impairment in mid-life (45–65 years old) was reported to be significantly associated with a 7% higher risk of dementia [1].

ARHL is physiologically caused by damage to the cochlear organs, including cochlear hair cells, spiral ganglion neurons, and the stria vascularis. It is accompanied by dysfunction of the central nervous system related to a decline in auditory information. Various studies at the cellular level have indicated that the production of reactive oxygen species (ROS) derived from mitochondrial dysfunction plays a key role in the development of ARHL [5,6]. However, the mechanism remains unclear.

The latest research indicates that the switching of mitochondrial metabolic substrates (glucose, amino acids, and fatty acids) plays an important role in aging [7] and that disruptions in the balance of mitochondrial metabolic substrates results in age-related metabolic diseases, such as diabetes, heart disease, and liver disease [8].

Cellular metabolism plays two main roles: breaking down molecules for energy (catabolism) and building cell components (anabolism). Although normal cells maintain this balance, an imbalance at the cellular level leads to mitochondrial DNA damage and accelerates cellular senescence [9], thus supporting the “mitochondrial theory of aging”.

Glycolysis is a central pathway in cellular energy metabolism that generates energy without requiring oxygen. In this process, one glucose molecule is broken down into two pyruvate molecules, producing small amounts of ATP and NADH. While ATP serves as an immediate energy source, NADH feeds into oxidative phosphorylation to generate additional ATP.

However, with aging, the efficiency of glycolysis may decline due to decreased enzyme activity and impaired glucose uptake. This reduction in glycolytic capacity can result in cellular energy deficits and increased stress, particularly in high-energy-demanding tissues such as the inner ear and brain. These metabolic disruptions may contribute to the functional decline associated with aging.

In lipid metabolism, stored triglycerides are broken down into fatty acids and glycerol. Fatty acids are converted into acetyl-CoA through β-oxidation in the mitochondria, thereby entering the citric acid cycle and producing ATP. This process is activated during energy-demanding situations, such as prolonged exercise or fasting.

However, aging is associated with impaired β-oxidation and mitochondrial dysfunction, which may result in the accumulation of fatty acids and lipotoxicity. These changes contribute to inflammation and oxidative stress, further promoting cellular senescence.

In amino acid metabolism, amino acids are used as an energy source, especially during nutrient deficiency. Amino acids are deaminated into ammonia and organic acids, which enter the citric acid cycle and generate energy. This energy supports growth, repair, and energy replenishment during shortages [10]. During aging, alterations in amino acid metabolism—such as a reduced availability of essential amino acids or imbalanced catabolism—can impair protein synthesis and autophagy. This may limit cellular repair mechanisms and exacerbate age-related cellular damage.

Glutamine (L-Gln) is the most abundant amino acid in the human body. Recent research has shown that L-Gln metabolism increases in senescent cells, which are cells that have stopped dividing and are associated with aging and age-related diseases. Glutaminase 1 (Gls1), an enzyme involved in L-Gln metabolism, plays a key role in this process. Inhibiting Gls1 with a compound such as Bis-2-(5-phenylacetamido-1,3,4-thiadiazol-2-yl) ethyl sulfide (BPTES) (a specific Gls1 inhibitor) has been shown to eliminate senescent cells and produce anti-aging effects in aged mice. This suggests that targeting L-Gln metabolism could be a potential strategy for combating aging and related cellular dysfunctions [11].

In this study, we analyzed the metabolic rates of glucose, amino acids, and fatty acids in cochlear cells, namely, House Ear Institute-Organ of Corti 1 (HEI-OC1) cells, and determined which substrate plays a key role in auditory cellular senescence. To clarify the metabolic characteristics of auditory cells under stress, this study examined substrate-specific differences in mitochondrial respiration using HEI-OC1 cells.

## 2. Materials and Methods

### 2.1. Cell Culture and Culture Conditions

House Ear Institute-Organ of Corti1 (HEI-OC1) cells are derived from the auditory organ of the Immortomouse™ (Charles River Laboratories, Wilmington, MA, USA), a transgenic mouse that carries a temperature-sensitive mutant of the SV40 large T-antigen gene under the control of an interferon-γ-inducible promoter [12]. These cells serve as a common progenitor for sensory and supporting cells of the organ of Corti. HEI-OC1 cells, kindly provided by Professor F. Kalinec (UCLA, Los Angeles, CA, USA), were cultured in high-glucose Dulbecco’s Eagle Medium (DMEM) (Life Technologies, Inc., Grand Island, NY, USA) supplemented with 10% fetal bovine serum (FBS) (Life Technologies, Inc., NY, USA) and 0.06% penicillin (Nacalai Tesque, Kyoto, Japan). The cells were incubated at 33 °C with 10% CO_2_ under permissive conditions. HEI-OC1 cells used in this study were maintained by weekly subculture from August 2022, and experiments were performed using cells between passages 55 and 95. All assays were conducted using cell cultures with ≥90% viability, as determined by trypan blue exclusion, to ensure the reliability and reproducibility of the results.

### 2.2. Reagents and Chemicals

The following reagents were obtained from Nacalai Tesque (Kyoto, Japan): 5.0 g/L trypsin/5.3 mmol/L EDTA solution (35556-44), 100 mM sodium pyruvate solution (100×) (06977-34), 200 mmol/L L-alanyl-L-glutamine solution (100×) (04260-64), 1 mol/L HEPES buffer solution (17557-94), palmitic acid sodium salt (25919-62), and antibiotic–antimycotic mixed stock solution (100×, stabilized) (09366-44).

The following reagents were purchased from FUJIFILM Wako Pure Chemical Corporation (Osaka, Japan): MEM essential amino acid solution (50×) (132-15641), MEM non-essential amino acid solution (100×) (139-15651), D-PBS (-) (045-29795), D-MEM (high glucose) (044-32955), D (+)-glucose (041-31165), disodium hydrogen phosphate dodecahydrate (196-02835), guaranteed reagent (074-00505), and L-phenylalanine (P5482-25G).

The fatty acid oxidation detection reagent (FAO Blue, FDV-0033) was obtained from Funakoshi (Tokyo, Japan).

The substrates tested included pyruvate; D-glucose (D-Glu); and amino acids, specifically L-Gln, L-glutamate (L-Glu), and mixtures of essential amino acids (EAA) and non-essential amino acids (NEAA). Although L-Gln and L-Glu are classified as non-essential amino acids, they are not included in the commercially available MEM NEAA solution and were supplemented separately. The EAA mixture included amino acids such as L-Arginine, L-Cysteine, L-Histidine, L-Isoleucine, L-Leucine, L-Lysine, L-Methionine, L-Phenylalanine, L-Threonine, L-Tryptophan, L-Tyrosine, and L-Valine, while the NEAA mixture included L-Alanine, L-Asparagine, L-Aspartic acid, Glycine, L-Proline, and L-Serine. Fatty acids such as palmitic acid (saturated) and oleic acid (unsaturated) were also measured.

### 2.3. Extracellular Flux Analysis

Cells were seeded in 24-well microplates of an XF24 Extracellular Flux Analyzer (100777-004; Seahorse Bioscience, Billerica, MA, USA), with four wells left blank, following the Seahorse XF24 User’s Manual [13].

After rinsing the cells twice with Dulbecco’s PBS (D-PBS; 14249-24, 14249-95, Nacalai Tesque, Kyoto, Japan; 045-29795, 049-29793, FUJIFILM Wako Pure Chemical Corporation, Osaka, Japan) and once with XF assay medium, they were resuspended in 675 µL of XF assay medium (DMEM without NaHCO_3_; 103335-100, Seahorse Bioscience, Billerica, MA, USA) and supplemented with 25 mM D-glucose and 1 mM sodium pyruvate (06977-34, Nacalai Tesque, Kyoto, Japan). Substrate concentrations were determined based on previous reports and Seahorse XF assay guidelines to ensure optimal mitochondrial response.

The concentrations of glucose and pyruvate were based on the standard culture conditions for HEI-OC1 cells. Glucose is commonly supplied in culture media at either 100 mg/dL (5.6 mM) or 450 mg/dL (25 mM), with the latter being used in the standard medium for HEI-OC1 cells and adopted in this study. Pyruvate was also used at 1 mM, in accordance with its concentration in the standard culture medium [14]. The concentrations of glucose, pyruvate, and L-Gln were matched to those typically used in the standard HEI-OC1 culture medium. Fatty acid concentrations were selected based on the most commonly reported conditions in previous studies. For inhibitor use, the concentration of BPTES was determined according to prior literature. The concentrations of oligomycin, FCCP, antimycin A, and rotenone were optimized through titration experiments to determine the most effective doses. (Details regarding substrate concentrations and inhibitor titration are provided in the Appendix A).

The plates were then equilibrated for 1 h at 33 °C in a non-CO_2_ incubator prior to OCR measurements. A specialized assay buffer was used for the measurements, and the pH was adjusted to 7.4 prior to the assay.

Sequential additions of mitochondrial inhibitors were performed during the assay: first, no additives were introduced to assess basal OCR; subsequently, oligomycin (final concentration: 0.6 µM) was added to inhibit ATP synthase and oxidative phosphorylation; FCCP (final concentration: 1 µM) was used to uncouple the proton gradient and induce maximum respiration; antimycin A (final concentration: 0.16 µM) was added to inhibit complex III; and, finally, rotenone (final concentration: 0.125 µM) was used to inhibit complex I. The OCR and extracellular acidification rate (ECAR) were measured following each injection [15].

The optimization of cell density and inhibitor concentrations was conducted prior to the analysis. Data were automatically calculated, recorded, and plotted using Seahorse XF24 software (version 1.8.1). All mitochondrial complex inhibitors were obtained as part of a kit from Seahorse Bioscience.

Following the assay, the cell numbers in each well were quantified to normalize the OCR values. In this experiment, the goal was to measure the OCR and the maximum OCR (MOCR) using various metabolic inhibitors [15].

In all graphs, OCR [%] is presented as a relative value normalized to the fourth time point, which corresponds to the baseline measurement immediately before the first drug injection. HEI-OC1 cells were seeded at a density of 3.15 × 10^4^ cells per well in XF24 cell culture microplates for extracellular flux analysis, unless otherwise specified. The specific substrate and inhibitor conditions used for each XF assay experiment are summarized in Table 1.

### 2.4. pH Measurement

The pH of the cell culture medium was determined using a portable pH meter (LAQUAtwin B-712, HORIBA, Kyoto, Japan) according to the manufacturer’s instructions. Calibration was performed using standard pH buffer solutions according to the manufacturer’s instructions.

For measurements using the fluorescent indicator SNARF-1 (Invitrogen, Thermo Fisher Scientific, Waltham, MA, USA), the fluorescent images were collected using a confocal microscope system (A1R; Nikon, Tokyo, Japan) with 60× (NA 0.95) Plan-Apo lens (excitation: 487.5 nm, dichroic mirror: BS 20/80, emission: 550–750 nm/resolution 10 nm/10 ch spectroscopy). The fluorescence emission was recorded at 590.1–608.1 (590) nm and 630–648.2 (630) nm. As the difference in the emission ratio (590/630) was too small to evaluate chronological changes, the signal intensity difference (590 nm–630 nm) was evaluated to quantify relative pH changes.

### 2.5. Mitochondrial Membrane Potential (MMP) Assay

The mitochondrial membrane potential (MMP) was measured using the mitochondrial fluorescent dye JC-1 (5,5′,6,6′-tetrachloro-1,1′,3,3′-tetraethyl benzimidazolyl carbocyanine iodide) (Biotium Inc., Fremont, CA, USA). JC-1 accumulates in mitochondria and forms either red-emitting JC-1 dimers at high membrane potentials or green-emitting JC-1 monomers at low membrane potentials. Cells were seeded in 24-well microplates (µ-Plate 24 Well ibiTreat, ibidi GmbH, Gräfelfing, Germany). After 30 min of incubation with 200 nM JC-1 (Biotium, USA), the medium was replaced with normal medium. The fluorescence emission ratio (red to green) was measured using an Infinite 200 Pro (TECAN JAPAN, Tokyo, Japan) with excitation at 535/9 nm (red) and 485/9 nm (green) and emission at 590/20 nm (red) and 535/20 nm (green). The ratio was calculated from multiple readings per well, with each well measured in a 3 × 3 square, integrated for 20 µs under 33 °C and 5% CO_2_. HEI-OC1 cells were seeded at a density of 13.13 × 10^4^ cells per well in µ-Plate 24 Well ibiTreat plates for JC-1 staining, unless otherwise specified.

### 2.6. Fatty Acid Assay

Fluorescence measurements were performed using an Infinite 200 Pro (TECAN JAPAN, Tokyo, Japan). Fatty acid oxidation activity was assessed using FAOBlue (Funakoshi, FDV-0033) following the manufacturer’s instructions. Etomoxir (Eto) was adjusted to a concentration of 40 μM, perhexiline (Per) to 10 μM, and BPTES to 3 μM before being administered to the cells in order to inhibit fatty acid oxidation, mitochondrial carnitine transport, and glutaminolysis, respectively.

The excitation and emission wavelengths for fluorescence measurement were set to Ex 405/9 nm and Em 460/20 nm, respectively. HEI-OC1 cells were seeded at a density of 19.20 × 10^4^ cells per well in µ-Plate 24 Well ibiTreat plates for FAOBlue staining, unless otherwise specified.

### 2.7. Statistical Analysis

All data are presented as mean ± standard deviation (SD) and were statistically evaluated using R version 4.5.1. Data were analyzed using Student’s *t*-test, and a *p*-value of <0.05 was considered statistically significant. Additionally, an analysis of variance (ANOVA) was performed for statistical analysis [16]. All pairwise *p*-values for multiple group comparisons are provided in the Appendix A.

Graphs for OCR quantification were plotted using Microsoft Excel.

## 3. Results

First, typical real-time changes in the OCR were measured using an XF24 Extracellular Flux Analyzer to evaluate mitochondrial respiration under basal and stressed conditions. The experimental procedure and definitions of each respiratory parameter are presented in Figure 1.

### 3.1. Glycolysis

To assess whether glucose-derived metabolites can sustain mitochondrial respiration, we performed OCR measurements after adding glucose, pyruvate, or UK5099 (a mitochondrial pyruvate carrier [MPC] inhibitor). These substrates and inhibitors were used to evaluate how efficiently glycolysis and mitochondrial pyruvate transport contribute to mitochondrial respiration (Figure 2).

The OCR gradually decreased over time due to substrate depletion. Glucose addition did not prevent this decline and showed no improvement in OCR compared to the control. In contrast, pyruvate supplementation maintained the OCR, indicating that glucose was not effectively glycolyzed and transferred into mitochondrial substrates under these conditions. This demonstrates that glycolytic flux or mitochondrial pyruvate import was insufficient, and that exogenous pyruvate bypassed this limitation and directly restored mitochondrial respiration.

When UK5099 was added, a further decline was observed compared to the control. Even with the addition of FCCP, metabolism did not recover. Pyruvate improved both the baseline and maximum OCRs compared to glucose. The MOCR of glucose was lower than that of pyruvate, suggesting that glucose alone was not efficiently converted into mitochondrial substrates under these conditions. This implies that the glycolytic flux was limited, possibly due to a regulatory bottleneck in the glycolysis pathway, and that exogenous pyruvate bypassed this limitation by directly entering the mitochondria, leading to greater OCR recovery.

### 3.2. Amino Acids

Several amino acids, including L-Gln and L-Glu, were added during the assay to evaluate their effects on cellular metabolic activity. As there are many types of amino acids, we started with a screening approach. As representative groups, we administered MEM EAA (L-Arginine, L-Cysteine, L-Histidine, L-Isoleucine, L-Leucine, L-Lysine, L-Methionine, L-Phenylalanine, L-Threonine, L-Tryptophan, L-Tyrosine, and L-Valine) and MEM NEAA (L-Alanine, L-Asparagine, L-Aspartic acid, Glycine, L-Proline, and L-Serine), which are commonly used in cell culture.

Among the amino acid substrates, only L-Gln significantly increased the baseline oxygen consumption rate (OCR), whereas EAA and NEAA showed no significant changes (Figure 3A).

Furthermore, following FCCP administration after L-Gln treatment, the maximum OCR (MOCR) was significantly elevated compared to EAA and NEAA treatments (Figure 3A). Since L-Gln is metabolized to L-Glu in mitochondria, we directly compared the OCR changes induced by L-Gln and L-Glu (Figure 3B). The baseline OCR was significantly increased by L-Gln, whereas no change was observed with L-Glu. These results indicate that L-Gln enhances mitochondrial activity.

The effect of Gls1 on oxygen consumption was investigated using the Gls1 inhibitor BPTES. Treatment with 3 µM BPTES, a concentration commonly used in previous studies, did not suppress the increase in OCR induced by L-Gln (Figure 4A). Increasing the BPTES concentration up to 40 µM also failed to alter OCR (Figure 4B).

Furthermore, the effect of increasing L-Gln concentrations on MOCR was assessed across multiple concentration ranges. Within each range (20, 40, and 80 mM; 5, 10, and 20 mM; 1.25, 2.5, and 5 mM), no significant differences were observed among the different concentrations. There was a slight tendency toward a lower increase in OCR at the 1.25 mM concentration. However, in all ranges, each concentration resulted in a significantly higher MOCR compared to the 0 mM control (Figure 5A–C).

As L-Gln metabolism could potentially affect cellular and extracellular pH levels, additional experiments were performed to investigate whether pH changes contributed to the observed alterations in the OCR.

The automatic pH measurement does not measure acidic pHs well. The recommended measuring range for pH is from 6.5 to 8.0, and if measurements are taken above this range, it is difficult to obtain accurate pH readings because they deviate from the linear line; thus, manual measurement was used to confirm the results. As shown in Figure 6C, manual pH measurements indicated that the lower detection limit was approximately pH 5.5. Only in the final section of the HCl-only treatment group did the pH fall below this limit, making statistical analysis for that section unfeasible. In contrast, the pH values in all other groups remained within the measurable range, allowing for rigorous statistical comparisons. The results of the ANOVA are provided in the Appendix A. This limitation should be taken into account when interpreting the data in Figure 6A,B.

As a result, no change in the OCR was observed when the pH increased alone; however, the OCR tended to increase further when pH increased in addition to L-Gln loading, suggesting synergistic effects. No increase in the OCR was observed with only an increase in pH. In addition, L-Gln plus a decrease in pH tended to significantly decrease the OCR.

Next, the intracellular pH was measured by using SNARF-1 (Seminaphthorhodafluor-1), allowing for the real-time monitoring of the pH changes in the cells. From the fluorescence ratio, it was determined that the intracellular pH decreased after L-Gln injection (Figure 7).

### 3.3. Fatty Acids

To examine the fatty acid metabolic pathway, both saturated and unsaturated fatty acids were administered to the cells to evaluate their effects on cellular metabolism and to determine how these different types of fatty acids influence metabolic activity. Both PA (palmitic acid, a saturated fatty acid) and OA (oleic acid, an unsaturated fatty acid) improved the baseline and maximum OCRs (Figure 8).

To directly assess fatty acid oxidation activity, fatty acid metabolism was evaluated using FAOBlue, a specific probe for fatty acid oxidation. A decrease in fatty acid metabolism was observed only in the group treated with Eto alone (Figure 9).

The baseline OCR was elevated in all fatty acid-treated groups compared to the BSA control, although the increase was smaller than that observed with L-Gln supplementation.

Similar findings were observed after FCCP administration; however, the maximum OCR increases induced by fatty acids were greater than those induced by L-Gln. Among the fatty acids, the OCR increase with PA was greater than that with OA.

We investigated the effects of Eto and perhexiline (Per) on the improvement in the OCR.

Based on the reduction in fatty acid metabolism observed with Eto treatment, further experiments were conducted to elucidate the effects of fatty acid oxidation inhibitors on the OCR under saturated fatty acid supplementation.

Per inhibits an enzyme called carnitine palmitoyl transferase (Cpt) in mitochondria. Specifically, it primarily inhibits carnitine palmitoyl transferase 1 (Cpt1) reversibly, with minimal effects on carnitine palmitoyl transferase 2 (Cpt2). In contrast, Eto is a selective irreversible inhibitor of Cpt1.

Therefore, the significant decrease in fatty acid metabolism observed in the Eto-treated group can be attributed to the near-complete inhibition of Cpt1, leading to a substantial suppression of β-oxidation in the mitochondria. As perhexiline reversibly inhibits Cpt1, fatty acid metabolism may have been partially maintained, possibly due to compensatory metabolic pathways.

These compounds reduce the transport of fatty acids into the mitochondria by inhibiting Cpt1, thereby suppressing fatty acid metabolism (β-oxidation). As a result, fatty acid utilization is suppressed, potentially leading to increased glucose utilization.

In the inhibitor pre-treatment experiments, the PA + Per group always exhibited an increased OCR compared to the PA group, whereas the PA + Eto group had a lower OCR than the PA group before FCCP treatment but an increased OCR after FCCP treatment (Figure 10A). In the co-administration experiments, the results were similar before FCCP administration. After FCCP administration, the OCR was particularly increased, especially in the PA + Eto group (Figure 10B).

The results of this experiment showed that both drugs significantly improved the MOCR in the group supplemented with saturated fatty acids.

Given the changes in oxygen consumption with fatty acid supplementation and inhibitor treatment, the effects on mitochondrial membrane potential were subsequently evaluated.

A decrease in membrane potential was observed at 30 min in the groups treated with BSA, OA, or L-Gln (Figure 11). In contrast, when saturated fatty acids were added, no changes in membrane potential were observed, not even after 30 min.

## 4. Discussion

In this study, we demonstrated that HEI-OC1 cells possess limited metabolic flexibility when relying on glucose under conditions of acute energy demand, despite glucose being traditionally regarded as the primary energy source [17]. Instead, we found that pyruvate and L-Gln more effectively supported mitochondrial respiration in the short term, while fatty acids increased OCR without contributing substantially to energy production. This observation is consistent with a mitochondrial uncoupling effect, in which oxygen consumption increases independently of ATP synthesis. Uncoupling typically results from proton leakage across the inner mitochondrial membrane, bypassing ATP synthase. Although OCR rises, the efficiency of oxidative phosphorylation decreases, and less usable ATP is generated. As such, elevated OCR in response to fatty acids does not necessarily indicate enhanced energy availability for cochlear cells. As shown in Figure 2, HEI-OC1 cells did not show significant OCR recovery upon glucose addition, whereas pyruvate administration suppressed OCR decline. Inhibition by UK5099 further reduced OCR. These findings clearly demonstrate that glycolysis dependent on glucose is insufficient to sustain mitochondrial respiration under acute energy demand in HEI-OC1 cells. These findings suggest that alternative substrates may play a critical role in maintaining energy homeostasis in the inner ear, particularly under conditions of acute energy demand or substrate limitation. This may reflect the physiological characteristics of cochlear cells, such as their reliance on oxidative metabolism and their limited glycolytic capacity. These traits, while supporting auditory precision, may also render cochlear cells more vulnerable to metabolic stress.

In contrast, other metabolites, such as pyruvate, L-Gln, and fatty acids, could be used as alternative substrates to address sudden spikes in energy demand. These findings are consistent with existing knowledge that non-glucose substrates play a key role in supporting cellular function under stress or in situations of high demand, particularly when cellular metabolism shifts towards more oxidative processes (Figure 12) [18].

Notably, L-Glu is a well-established neurotransmitter in the inner ear [19], and HEI-OC1 cells may also express L-Glu receptors. Therefore, the observed increase in oxygen consumption or proliferation upon L-Glu addition may not be solely attributed to its role as a metabolic substrate but may also involve receptor-mediated activation. However, in the present study, L-Glu administration did not induce an increase in OCR, whereas L-Gln alone significantly elevated OCR (Figure 3B). These findings suggest that the increase in OCR in this system is not driven by neurotransmitter-mediated stimulation via L-Glu but rather by metabolic processes involving glutaminase. Specifically, as shown in Figure 3A, only L-Gln significantly increased OCR, while other essential and non-essential amino acids had no notable effect. Figure 3B further confirms that L-Glu did not elevate OCR, supporting the conclusion that OCR increase is due to L-Gln metabolism rather than L-Glu receptor activation. Our data showed that L-Gln, but not L-Glu, increased both basal OCR and MOCR in HEI-OC1 cells. Under conditions where mitochondrial pyruvate utilization is limited, L-Gln can sustain TCA cycle flux by supplying both oxaloacetate and acetyl-CoA via glutaminase and GDH-dependent pathways, whereas exogenous L-Glu does not equivalently substitute unless routed through glutaminolysis [20]. This paradigm—glutamine’s metabolic flexibility under stress—has been demonstrated when the mitochondrial pyruvate carrier is inhibited, where GDH-dependent rerouting of L-Gln carbons maintains respiration and survival [20]. The concordance between that mechanism and our HEI-OC1 phenotype supports the interpretation that L-Gln, rather than L-Glu, functions as the more effective substrate to acutely drive mitochondrial respiration in cochlear lineage cells. Potential contributing factors include differential transport/compartmentalization of L-Glu, regulation of GLS/GDH, and the need for dual provision of OAA and acetyl-CoA. Similarly, fatty acids may also act through signaling pathways, including nuclear receptor activation [21], which complicates the interpretation of their metabolic effects. Further studies are required to distinguish these metabolic and signaling effects.

As shown in Figure 4A,B, BPTES-mediated inhibition of Gls1 at concentrations ranging from 0 to 40 µM did not suppress L-Gln-induced OCR increase. This suggests the presence of Gls1-independent metabolic pathways or compensatory effects by other glutaminases in HEI-OC1 cells. These results indicate that Gls1 is not the dominant contributor to L-Gln metabolism in these cells. Therefore, we considered the potential involvement of Gls2, another glutaminase isoform with distinct regulatory functions. In contrast, cancer cells with high Gls2 expression exhibit limited effects upon Gls1 inhibition, suggesting that Gls2 may compensate for metabolic flexibility [22]. This may indicate the involvement of Gls1-independent pathways of L-Gln metabolism, such as Gls2-mediated transamination, or reflect the insufficient inhibitory effect of BPTES in inner ear cells.

We speculate that BPTES, a Gls1 inhibitor, might not be effective in mitigating hearing loss caused by mitochondrial dysfunction and impaired metabolic flexibility in cochlear cells because Gls1 did not play a central role in energy metabolism in these cells. However, the observed links between the glucose and L-Gln pathways suggest that Gls2 may be a key player in this process, particularly under conditions where L-Gln is converted to intermediates that support mitochondrial energy production [23]. While the lack of specific Gls2 inhibitors limited our ability to directly test this hypothesis, the potential involvement of Gls2 warrants further investigation. The current inability to selectively inhibit Gls2 might highlight the need for more targeted tools to investigate the role of this enzyme in the cellular metabolism of the cochlea. Gls2 is a known transcriptional target of the p53 pathway, and it has been shown to play a crucial role in maintaining mitochondrial function and antioxidant capacity under oxidative stress conditions [24]. Given that cochlear cells are highly susceptible to oxidative damage due to their elevated metabolic activity, the activation of Gls2 may enhance cellular resistance to stress. Thus, it is plausible that Gls2 contributes to the maintenance of inner ear homeostasis by supporting mitochondrial integrity and mitigating oxidative stress-induced damage.

Our findings suggest that saturated fatty acids may serve as more efficient energy substrates than unsaturated fatty acids, which represents an interesting avenue for further research [22]. Furthermore, a recent study demonstrated that a high-fat diet enriched in saturated fatty acids significantly protected against progressive hearing loss in C57BL/6J mice, a strain highly susceptible to age-related hearing loss [25]. Our findings on the metabolic activation by palmitate may provide a mechanistic basis for this observation, suggesting that specific saturated fatty acids could support cochlear bioenergetics and confer resilience to metabolic stress in aging auditory systems. This differential utilization of fatty acids may be related to the specific bioenergetic needs of cochlear cells, which may prioritize certain fatty acids based on their oxidative capacity and contribution to ATP production. In addition, mitochondrial uncoupling induced by fatty acids may influence the metabolic stress response and vulnerability of cochlear cells during aging. Investigating the interplay between fatty acid metabolism and aging-related pathways in the cochlea may provide further insight into the pathogenesis of age-related hearing loss.

Although there was no significant difference in MOCR among varying L-Gln concentrations (Figure 5A–C), all concentrations significantly increased MOCR compared to 0 mM. Regarding pH, only the HCl-only group fell below the measurable pH limit (~5.5) in Figure 6C, while other groups remained within analyzable range. Figure 6A,B show that pH elevation alone did not notably alter OCR, but a combination of lowered pH and L-Gln tended to suppress OCR. Furthermore, intracellular acidification following L-Gln treatment was confirmed by SNARF-1 measurements (Figure 7). Ammonia, produced through the deamination of L-Gln by glutaminase, likely contributed to the increase in intracellular pH by binding with protons inside the cell [11,26].

Protons may originate from intracellular metabolism and may be extruded to maintain pH [27], thereby acidifying the extracellular space. Alternatively, extracellular acidification may lead to a reduction in intracellular pH when buffering systems are overwhelmed [28]. The L-Gln concentration affects pH, leading to acidification. This may hinder ammonia neutralization, resulting in a hypermetabolic state.

Both saturated and unsaturated fatty acid treatments increased baseline and maximum OCRs, although the magnitude was less than that observed with L-Gln (Figure 8). Using FAOBlue as a marker of fatty acid oxidation, significant suppression was observed only with Eto alone (Figure 9). In combination with inhibitors, PA + Per consistently showed higher OCR, while PA + Eto showed temporary OCR reduction before FCCP, with recovery afterwards (Figure 10A,B). Membrane potential measurements indicated no change after 30 min with PA treatment, while OA, L-Gln, and BSA showed a decreasing trend (Figure 11).

The reason why saturated or unsaturated fatty acids (OA) did not cause changes in membrane potential may be related to their effect on membrane fluidity. PA, with its straight-chain structure, tends to arrange more orderly in the lipid bilayer, making the membrane stiffer. This stiffening reduces membrane permeability and suppresses the activity of ion channels and pumps, leading to a more stable membrane potential. Conversely, unsaturated fatty acids (OA), due to their double bonds, introduce “kinks” in the fatty acid chains, increasing membrane fluidity. This enhances membrane permeability, promoting ion movement and more readily causing changes in membrane potential [29].

Thus, saturated fatty acids (PA) may suppress membrane potential changes by stabilizing the membrane, while unsaturated fatty acids (OA) likely have the opposite effect. As ion channels and membrane potential dynamics are crucial for cochlear hair cell function, alterations in membrane fluidity could affect mechanotransduction and auditory signaling. An increased membrane stiffness might impair the ion flux necessary for hair cell depolarization, potentially contributing to age-related or noise-induced hearing loss [30,31].

Furthermore, membrane lipid metabolism is related more to membrane function than to energy storage, which could explain why the saturated fatty acid-loaded group showed a greater OCR increase than the L-Gln-loaded group [32]. Given the high metabolic demands of cochlear cells, disruptions in lipid-mediated mitochondrial activity might exacerbate oxidative stress, further implicating fatty acid metabolism in cochlear dysfunction and hearing loss.

Overall, these findings highlight the differential metabolic substrate utilization in HEI-OC1 cells, emphasizing the limited reliance on glucose and the significant roles of L-Gln, pyruvate, and condition-dependent fatty acid metabolism (Figure 2, Figure 3, Figure 8, Figure 9, Figure 10 and Figure 11).

Further investigation into Gls2 function and cochlear fatty acid metabolism may offer promising strategies for preserving cochlear function. Notably, saturated fatty acids could play a critical role in cellular bioenergetics and protection against age-related hearing loss.

This study has several limitations. First, we did not include a BPTES-only condition, which prevents the exclusion of potential direct effects of BPTES on OCR. Second, we did not assess senescence-related markers such as SA-β-galactosidase. Investigating the relationship between substrate metabolism and cellular senescence in cochlear cells may provide further mechanistic insight. Third, this study used the immortalized HEI-OC1 cell line, which does not fully replicate the physiological complexity of the cochlea, including multicellular interactions, vascularization, and systemic regulation. Therefore, caution should be exercised when translating these findings into the physiological context. To enhance the physiological relevance of our findings, future studies using primary cochlear cells or in vivo animal models, such as aged mice, are warranted. Moreover, mitochondrial uncoupling by fatty acids may influence the vulnerability of cochlear cells to metabolic stress during aging. Exploring this link could provide new perspectives on age-related hearing loss and cochlear energy metabolism.

## 5. Conclusions

Pyruvate and L-Gln are more effective for short-term energy needs, whereas fatty acids influence the mitochondrial OCR but may play a limited role as an energy source in inner ear cells. However, glucose cannot rapidly meet sudden energy demands while it supports basic cell functions. Investigating the role of Gls2 with targeted inhibitors could provide deeper insights into metabolic flexibility in cochlear dysfunction and hearing loss.

## Figures and Tables

**Figure 1 biology-14-01118-f001:**
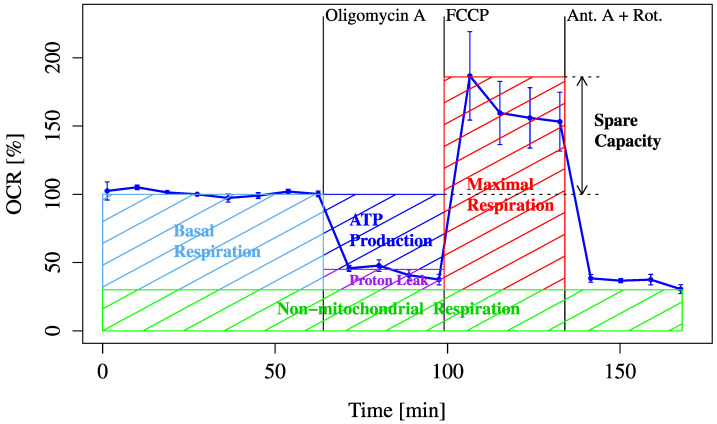
Schematic representation of typical OCR measurement workflow. This figure illustrates the typical sequence of mitochondrial OCR measurement using extracellular flux analysis. Basal respiration is measured first, followed by oligomycin to assess ATP-linked respiration and FCCP to determine maximum respiratory capacity. Subsequent substrate and inhibitor loading steps are used to evaluate metabolic flexibility.

**Figure 2 biology-14-01118-f002:**
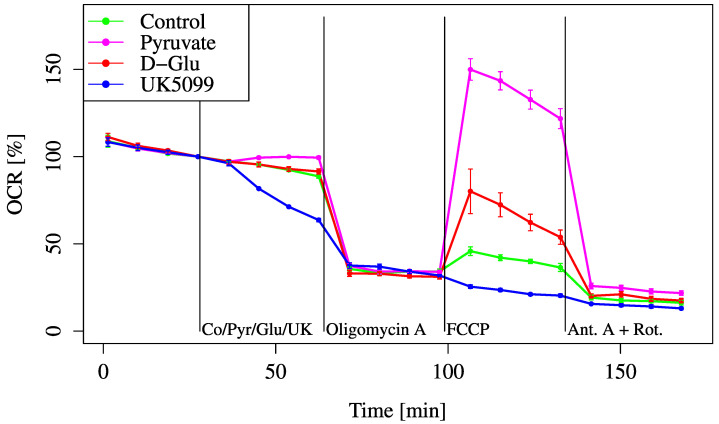
OCR responses to pyruvate, glucose, or UK5099. Cells were incubated in D-glucose/pyruvate-free XF assay medium to establish baseline respiration (time = 0). Control (Co) (medium only), Pyruvate (Pyr) (2 mM, final), D-glucose (Glu) (24.7 mM, final), or UK5099 (UK) (10 μM, final) was then added to evaluate substrate-dependent OCR changes. All data are presented as mean ± SD; *n* = 5 per group.

**Figure 3 biology-14-01118-f003:**
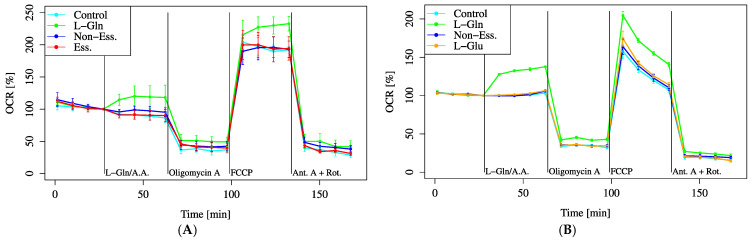
OCR responses to amino acid substrates. Cells were incubated in XF assay medium containing pyruvate (2 mM) and D-glucose (25 mM) to establish baseline respiration (time = 0). Due to instrument constraints, only three substrates could be compared per assay. Therefore, (**A**,**B**) represent independent experiments with partially overlapping conditions. (**A**) OCR changes were measured after the addition of L-Gln (4 mM, final), essential amino acids (Ess.) (1:25 dilution, final), or non-essential amino acids (Non-Ess.) (1:50 dilution, final). (**B**) OCR was assessed with L-Gln (4 mM, final), L-glutamate (L-Glu) (0.25 mM, final), or non-essential amino acids (Non-Ess.) (1:50 dilution, final). The L-glutamate concentration was set to match the typical level in standard HEI-OC1 culture medium; however, this relatively low concentration may have limited the observed increase in OCR. All data are presented as mean ± SD; *n* = 5 per group.

**Figure 4 biology-14-01118-f004:**
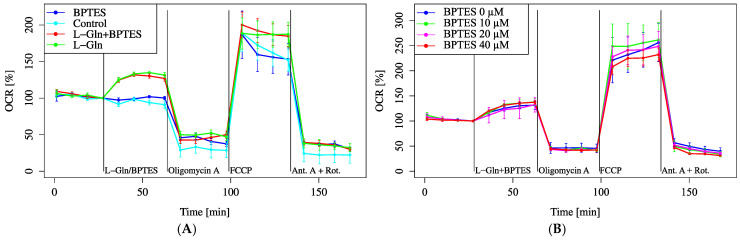
OCR modulation by BPTES. Cells were incubated in XF assay medium containing pyruvate (2 mM) and D-glucose (25 mM) to establish baseline respiration (time = 0). (**A**) OCR was measured after the addition of L-Gln (4 mM, final), BPTES (3 μM, final), or both. For the first drug injection (labeled as L-Gln/BPTES), the control received no additional treatment, whereas each treatment group received the indicated compounds. (**B**) OCR was evaluated with increasing concentrations of BPTES. All data are presented as mean ± SD; *n* = 5 per group.

**Figure 5 biology-14-01118-f005:**
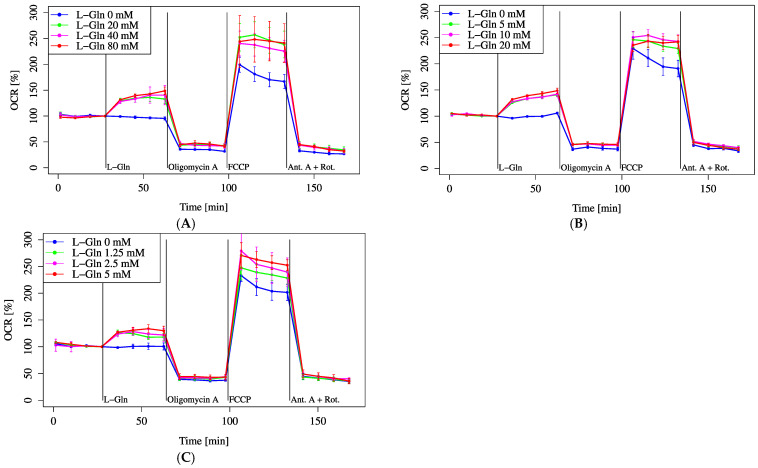
OCR responses to graded glutamine concentrations. Cells were incubated in XF assay medium with pyruvate (2 mM) and D-glucose (25 mM) to establish baseline respiration (time = 0). OCR was monitored following the addition of L-Gln across three concentration ranges: (**A**) 0–80 mM, (**B**) 0–20 mM, and (**C**) 0–5 mM. All data are presented as mean ± SD; *n* = 5 per group.

**Figure 6 biology-14-01118-f006:**
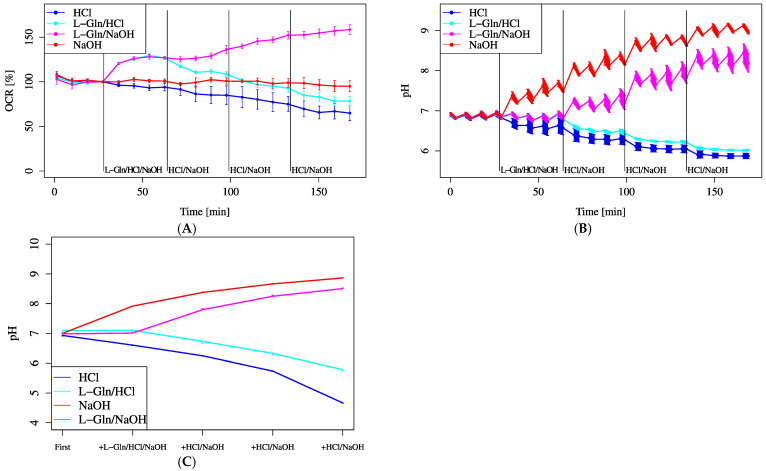
Effect of pH shifts on OCR. Cells were incubated in XF assay medium containing pyruvate (2 mM) and D-glucose (25 mM). (**A**) OCR changes and (**B**) pH shifts were monitored following stepwise pH modification using HCl or NaOH, each added at base 675 µL + 40 mML-Gln 75 µL + 2.5 mMHCl, 5 mM NaOH 75 µL × 3 (L-Gln/HCl or L-Gln/NaOH)/base 675 µL + 2.5 mMHCl, or 5 mM NaOH 75 µL × 4 (HCl or NaOH). (**C**) Average pH change per treatment was manually confirmed. L-Gln was treated at 4 mM (final). Data are shown as mean ± SD ((**A**,**B**): *n* = 5; (**C**): *n* = 3). Unpaired two-tailed *t*-tests were performed to evaluate statistical differences between the HCl and L-Gln/HCl groups and between the NaOH and L-Gln/NaOH groups. No statistically significant differences were observed in either comparison (HCl vs. L-Gln/HCl, *p* = 0.087; NaOH vs. L-Gln/NaOH, *p* = 0.086).

**Figure 7 biology-14-01118-f007:**
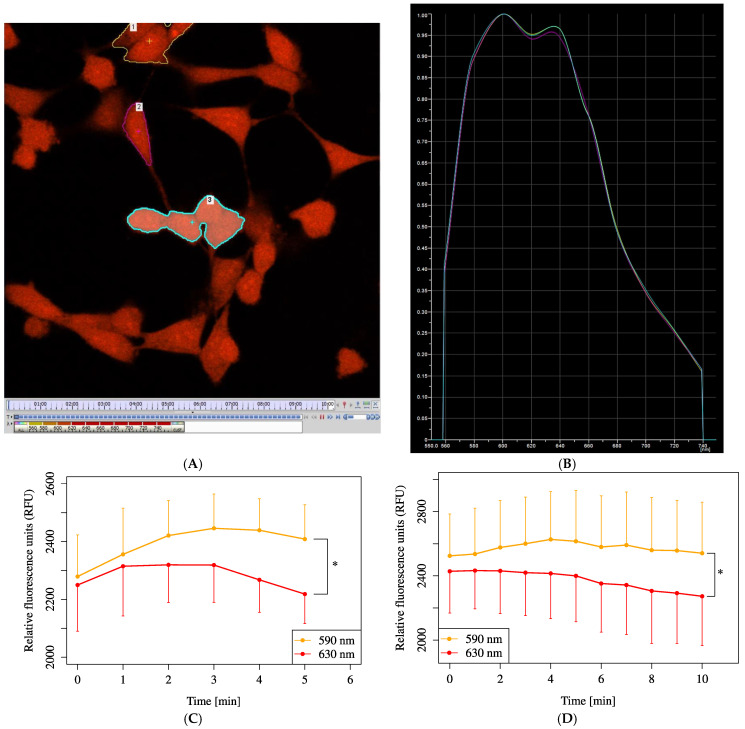
Intracellular pH changes measured using SNARF-1. Cells were initially incubated in XF assay medium with pyruvate (2 mM) and D-glucose (25 mM) to establish the baseline (time = 0). (**A**) Fluorescence image of HEI-OC1 cells stained with SNARF-1 to visualize intracellular pH. Three representative cells were manually selected from the video data, and the fluorescence intensity over time was extracted. (**B**) Time-lapse image extracted from the SNARF-1 fluorescence video. Due to limitations of the imaging system, dynamic pH measurements could not be acquired directly in a format matching (**C**,**D**). (**C**) Changes in SNARF-1 relative fluorescence units following the addition of L-Gln (200 µL) to base medium (500 µL) at 30 s (*p* < 0.01). (**D**) Changes in SNARF-1 relative fluorescence units following the addition of only L-Gln (200 µL) at 30 s (*p* < 0.01). All data are presented as mean ± SD. *n* = 3 per group. Statistical comparison was performed using a two-tailed unpaired *t*-test. Values of *: *p* < 0.05 were considered statistically significant.

**Figure 8 biology-14-01118-f008:**
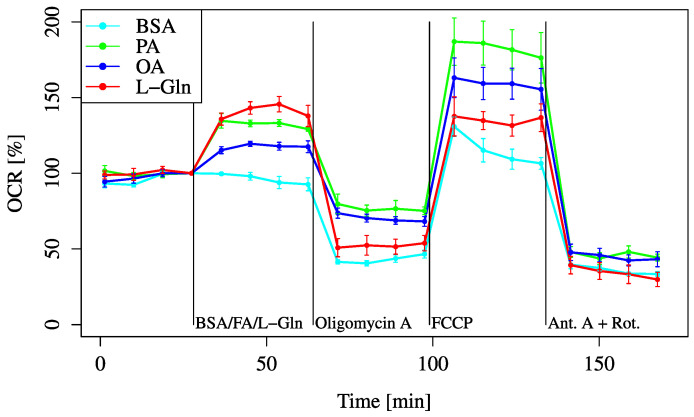
OCR responses to fatty acid (FA) substrates. Cells were incubated in XF assay medium containing pyruvate (2 mM), D-glucose (24.7 mM), and L-carnitine (775 μM) to establish the baseline (time = 0). OCR was measured after treatment with BSA (0.034 mM, final), L-Gln (4 mM, final), palmitic acid (PA, 0.2 mM, final), or oleic acid (OA, 0.2 mM, final). All data are shown as mean ± SD; *n* = 5 per group.

**Figure 9 biology-14-01118-f009:**
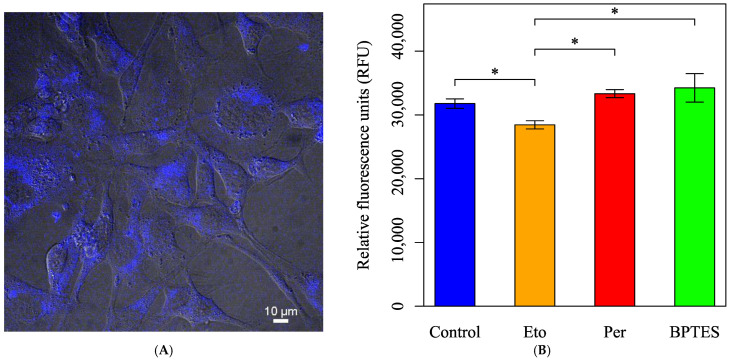
Evaluation of fatty acid metabolism using the FAOBlue dye. Cells were initially incubated in normal culture medium with pyruvate, D-glucose, and L-Gln. (**A**) The fluorescence image of the HEI-OC1 cells stained with FAOBlue. (**B**) Cells were treated with Eto (40 µM, final), Per (10 µM, final), or BPTES (3 µM, final), an inhibitor of glutaminase, for 30 min; were stained with FAOBlue in HBSS (+) for 30 min; and were measured to compare the effect of fatty acid oxidation activity. All data are presented as mean ± SD. *n* = 6 per group. Statistical comparisons were performed using two-tailed unpaired *t*-tests. Significant differences were observed between the following groups: Control vs. Eto (*: *p* = 0.015), Control vs. Per (*: *p* = 0.038), Eto vs. Per (*: *p* = 0.010). *p* values less than 0.05 were considered statistically significant.

**Figure 10 biology-14-01118-f010:**
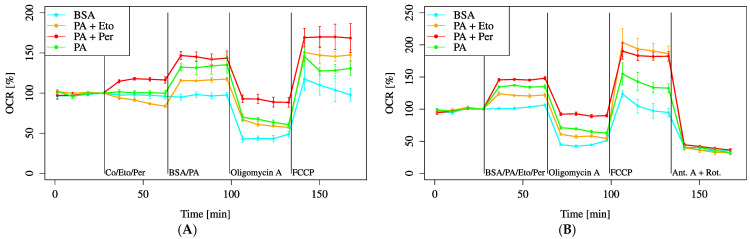
Comparison of OCR recovery with etomoxir (Eto) and perhexiline (Per). Cells were incubated in an XF assay medium containing pyruvate (2 mM), D-glucose (24.7 mM), and L-carnitine (775 μM) to establish the baseline (time = 0). (**A**) Eto (40 μM, final) and Per (10 μM, final) were added sequentially, followed by BSA. (**B**) Eto, Per, and BSA were simultaneously added. All data are shown as mean ± SD; *n* = 5 per group.

**Figure 11 biology-14-01118-f011:**
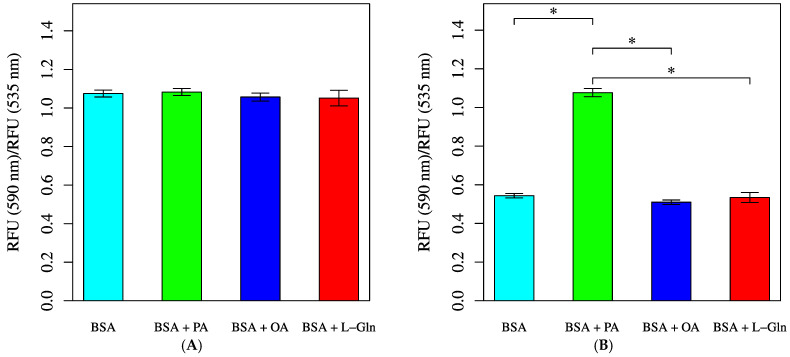
Mitochondrial membrane potential measured by JC-1 staining. Cells were incubated in an XF assay medium containing pyruvate (2 mM), D-glucose (24.7 mM), and L-carnitine (775 μM) to establish the baseline (time = 0). Mean of fluorescence intensity ratios were measured after treatment with BSA (0.034 mM, final), L-Gln (4 mM, final), BSA + palmitic acid (PA, 0.2 mM, final), or BSA + oleic acid (OA, 0.2 mM, final). (**A**) JC-1 fluorescence intensity ratio at baseline (*p* = 0.2203). (**B**) JC-1 fluorescence intensity ratio following BSA/PA/OA/L-Gln injection and 30-min incubation at 33 °C. Statistical significance was assessed using ANOVA. All data are presented as mean ± SD; *n* = 6 per group; *: *p* < 0.001.

**Figure 12 biology-14-01118-f012:**
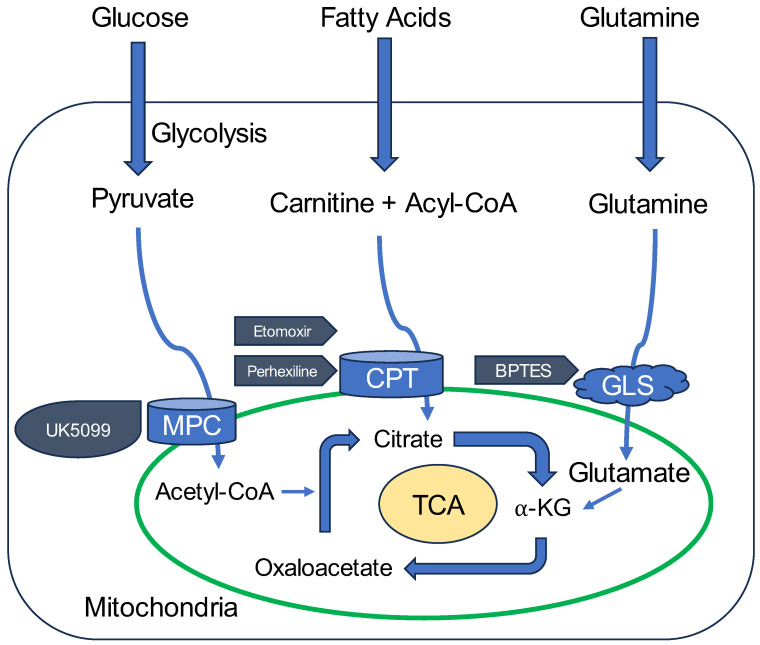
Conceptual summary of mitochondrial metabolic pathways and regulation by inhibitors. This schematic illustrates key mitochondrial metabolic pathways examined in this study, including glucose, amino acid (particularly L-Gln), and fatty acid oxidation. The figure integrates findings from OCR analyses and highlights the roles of specific metabolic inhibitors. UK5099 inhibits mitochondrial pyruvate carrier (MPC), thereby blocking the entry of pyruvate into mitochondria. Eto irreversibly inhibits carnitine palmitoyl transferase 1 (Cpt1), suppressing mitochondrial β-oxidation of fatty acids. Perhexiline reversibly inhibits Cpt1, leading to partial suppression of fatty acid oxidation. BPTES targets Gls1, an enzyme involved in L-Gln metabolism, though its inhibition did not suppress L-Gln-induced OCR elevation in HEI-OC1 cells. MPC: mitochondrial pyruvate carrier, CPT: carnitine palmitoyltransferase, GLS: glutaminase, TCA: tricarboxylic acid cycle, α-KG: α-ketoglutaric acid.

**Table 1 biology-14-01118-t001:** Summary of substrate and inhibitor conditions for each XF assay experiment.

	Initial Medium (675 µL)	Port A (75 µL)	Port B (75 µL)	Port C (75 µL)	Port D (75 µL)
Figure 1	Same as Figure 4A (BPTES)				
Figure 2	XF assay medium (103335-100)	(Control)/ Sodium pyruvate (final 2 mM)/ D-Glucose (final 24.7 mM)/ UK5099 (final 10 μM)	Oligomycin A (final 0.6 µM)	FCCP (final 1 µM)	Antimycin A (final 0.16 µM) +Rotenone (final 0.125 µM)
Figure 3A	XF assay medium (103335-100) +25 mM D-Glucose +1 mM Sodium pyruvate	(Control)/ L-Gln (final 4 mM)/ Non-Ess. (final 1:50)/ Ess. (final 1:25)	Oligomycin A (final 0.6 µM)	FCCP (final 1 µM)	Antimycin A (final 0.16 µM) +Rotenone (final 0.125 µM)
Figure 3B	XF assay medium (103335-100) +25 mM D-Glucose +1 mM Sodium pyruvate	(Control)/ L-Gln (final 4 mM)/ Non-Ess. (final 1:50)/ L-Glu (final 0.25 mM)	Oligomycin A (final 0.6 µM)	FCCP (final 1 µM)	Antimycin A (final 0.16 µM) +Rotenone (final 0.125 µM)
Figure 4A	XF assay medium (103335-100) +25 mM D-Glucose +1 mM Sodium pyruvate	BPTES (final 3 μM)/ (Control)/ L-Gln (final 4 mM) + BPTES (final 3 μM)/ L-Gln (final 4 mM)	Oligomycin A (final 0.6 µM)	FCCP (final 1 µM)	Antimycin A (final 0.16 µM) +Rotenone (final 0.125 µM)
Figure 4B	XF assay medium (103335-100) +25 mM D-Glucose +1 mM Sodium pyruvate	L-Gln (final 4 mM)/ L-Gln (final 4 mM) + BPTES (final 10 μM)/ L-Gln (final 4 mM) + BPTES (final 20 μM)/ L-Gln (final 4 mM) + BPTES (final 40 μM)	Oligomycin A (final 0.6 µM)	FCCP (final 1 µM)	Antimycin A (final 0.16 µM) +Rotenone (final 0.125 µM)
Figure 5A	XF assay medium (103335-100) +25 mM D-Glucose +1 mM Sodium pyruvate	(Control)/ L-Gln (final 20 mM)/ L-Gln (final 40 mM)/ L-Gln (final 80 mM)	Oligomycin A (final 0.6 µM)	FCCP (final 1 µM)	Antimycin A (final 0.16 µM) +Rotenone (final 0.125 µM)
Figure 5B	XF assay medium (103335-100) +25 mM D-Glucose +1 mM Sodium pyruvate	(Control)/ L-Gln (final 5 mM)/ L-Gln (final 10 mM)/ L-Gln (final 20 mM)	Oligomycin A (final 0.6 µM)	FCCP (final 1 µM)	Antimycin A (final 0.16 µM) +Rotenone (final 0.125 µM)
Figure 5C	XF assay medium (103335-100) +25 mM D-Glucose +1 mM Sodium pyruvate	(Control)/ L-Gln (final 1.25 mM)/ L-Gln (final 2.5 mM)/ L-Gln (final 5 mM)	Oligomycin A (final 0.6 µM)	FCCP (final 1 µM)	Antimycin A (final 0.16 µM) +Rotenone (final 0.125 µM)
Figure 6A	XF assay medium (103335-100) +25 mM D-Glucose +1 mM Sodium pyruvate	HCl (inject 2.5 mM)/ L-Gln (inject 40 mM)/ L-Gln (inject 40 mM)/ NaOH (inject 5 mM)	HCl (inject 2.5 mM)/ HCl (inject 2.5 mM)/ NaOH (inject 5 mM)/ NaOH (inject 5 mM)	HCl (inject 2.5 mM)/ HCl (inject 2.5 mM)/ NaOH (inject 5 mM)/ NaOH (inject 5 mM)	HCl (inject 2.5 mM)/ HCl (inject 2.5 mM)/ NaOH (inject 5 mM)/ NaOH (inject 5 mM)
Figure 8	XF assay medium (103335-100) +Pyruvate (2 mM) +D-Glucose (24.7 mM) +L-carnitine (775 μM) adjusted to pH7 using 1M NaOH	BSA (final 0.034 mM)/ BSA + PA (final 0.2 mM)/ BSA + OA (final 0.2 mM)/ L-Gln (finak 4 mM)	Oligomycin A (final 1.2 µM)	FCCP (final 2 µM)	Antimycin A (final 0.32 µM) +Rotenone (final 0.25 µM)
Figure 10A	XF assay medium (103335-100) +Pyruvate (2 mM) +D-Glucose (24.7 mM) +L-carnitine (775 μM) adjusted to pH7 using 1M NaOH	(Control)/ (Control)/ Eto (final 40 μM)/ Per (final 10 μM)	BSA (final 0.034 mM)/ BSA + PA (final 0.2 mM)/ BSA + PA (final 0.2 mM)/BSA + PA (final 0.2 mM)	Oligomycin A (final 1.2 µM)	FCCP (final 2 µM)
Figure 10B	XF assay medium (103335-100) +Pyruvate (2 mM) +D-Glucose (24.7 mM) +L-carnitine (775 μM) adjusted to pH7 using 1M NaOH	BSA (final 0.034 mM)/ BSA + PA (final 0.2 mM)/ BSA + PA (final 0.2 mM) + Eto (final 40 μM)/ BSA + PA (final 0.2 mM) + Per (final 10 μM)	Oligomycin A (final 1.2 µM)	FCCP (final 2 µM)	Antimycin A (final 0.32 µM) +Rotenone (final 0.25 µM)

## Data Availability

The raw data supporting the conclusions of this article will be made available by the authors without undue reservation.

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
