# Peer review of "Rates of Mitochondrial Metabolism of Glucose, Amino Acids, and Fatty Acids by the HEI-OC1 Inner Ear Cell Line"

_biology, 2025, doi:10.3390/biology14091118_

Round 1

Reviewer 1 Report

Comments and Suggestions for Authors

This study focuses on the mitochondrial metabolic rates of glucose, amino acids, and fatty acids in the HEI-OC1 inner ear cell line. The main findings are that cochlear cells primarily rely on glucose for basal metabolism, while pyruvate and glutamine serve as key substrates for acute energy demands, with a limited role for fatty acids. The findings provide an important basis for exploring metabolic intervention strategies for ARHL. It is a detailed study, primarily employing biochemical methods. I believe the experiments are of good quality and warrant publication. I have a few suggestions that may help improve the manuscript's academic rigor, as outlined below.

The story of fatty acids is of particular interest to me. A recent study found that feeding C57BL/6J mice a high-fat diet, primarily composed of saturated fat, showed a surprising protective effect on their hearing (Zhang et al, 2023, The FASEB J). The high-fat diet virtually cured the signature early onset of progressive hearing loss in that particular strain. It appears to me that the findings in this manuscript could address the mechanism at the molecular level. I suggest the authors expand the discussion in that direction.

The use of "maximum OCR" and "maximal OCR" should be standardized to avoid confusion.

A detailed explanation of the "mitochondrial uncoupling effect" is needed. In the discussion section, it is advisable to supplement the explanation of how this effect influences ATP production efficiency, allowing readers to better understand its functional implications.

The conclusion section could further highlight the mechanistic association between metabolic substrates and ARHL to reinforce the study’s practical significance.

The experimental section does not specify the passage number of HEI-OC1 cells, which significantly impacts mitochondrial function. Adding this information is recommended.

In the amino acid metabolism experiments, including a separate group treated with a glutaminase inhibitor (e.g., BPTES) would help to confirm the specific role of glutamine metabolism.

To bridge the gap between the HEI-OC1 cell line and in vivo physiological conditions, validation using primary cochlear cells or animal models (e.g., aged mice) is recommended to improve the physiological relevance of the results.

The discussion could be expanded to explore the association between fatty acid uncoupling effects and cochlear cell senescence.

Including analyses of senescence-associated markers (e.g., β-galactosidase) in relation to substrate metabolism would enhance the mechanistic understanding of metabolism and hair cell senescence.

Author Response

Response to Reviewer #1

We sincerely thank Reviewer #1 for the thorough reading of our manuscript and the constructive comments. Below, we provide our point-by-point responses to each comment, strictly following the original suggestions.

Reviewer’s comment:
The story of fatty acids is of particular interest to me. A recent study found that feeding C57BL/6J mice a high-fat diet, primarily composed of saturated fat, showed a surprising protective effect on their hearing (Zhang et al, 2023, The FASEB J). The high-fat diet virtually cured the signature early onset of progressive hearing loss in that particular strain. It appears to me that the findings in this manuscript could address the mechanism at the molecular level. I suggest the authors expand the discussion in that direction.
Response:
Thank you for your valuable comment. The findings of Zhang et al. are highly interesting and potentially consistent with our observation of metabolic activation by palmitate. We have added a mechanistic discussion indicating that saturated fatty acids may contribute to mitochondrial energy supply in the inner ear and potentially exert protective effects in age-related hearing loss, consistent with the study by Zhang et al.

(Furthermore, a recent study demonstrated that a high-fat diet enriched in saturated fatty acids significantly protected against progressive hearing loss in C57BL/6J mice, a strain highly susceptible to age-related hearing loss (Zhang et al., 2023, The FASEB Journal). Our findings on the metabolic activation by palmitate may provide a mechanistic basis for this observation, suggesting that specific saturated fatty acids could support cochlear bioenergetics and confer resilience to metabolic stress in aging auditory systems[23].)
(Page.19,Line518-524)

Reviewer’s comment:
The use of "maximum OCR" and "maximal OCR" should be standardized to avoid confusion.
Response:
Thank you for pointing this out. All instances have been unified as "maximum OCR."

Reviewer’s comment:
A detailed explanation of the "mitochondrial uncoupling effect" is needed. In the discussion section, it is advisable to supplement the explanation of how this effect influences ATP production efficiency, allowing readers to better understand its functional implications.
Response:
In accordance with your suggestion, we have added an explanatory sentence regarding this point.

(This observation is consistent with a mitochondrial uncoupling effect, in which oxygen consumption increases independently of ATP synthesis. Uncoupling typically results from proton leakage across the inner mitochondrial membrane, bypassing ATP synthase. Although OCR rises, the efficiency of oxidative phosphorylation decreases, and less usable ATP is generated. As such, elevated OCR in response to fatty acids does not necessarily indicate enhanced energy availability for cochlear cells. As shown in Figure 2, HEI-OC1 cells did not show significant OCR recovery upon glucose addition, whereas pyruvate administration suppressed OCR decline. Inhibition by UK5099 further reduced OCR. These findings clearly demonstrate that glycolysis dependent on glucose is insufficient to sustain mitochondrial respiration under acute energy demand in HEI-OC1 cells. These findings suggest that alternative substrates may play a critical role in maintaining energy homeostasis in the inner ear, particularly under conditions of acute energy demand or substrate limitation.)

(Page.17,Line452-464)

Reviewer’s comment:
The conclusion section could further highlight the mechanistic association between metabolic substrates and ARHL to reinforce the study’s practical significance.
Response:
We appreciate the suggestion. However, as the precise mechanisms remain speculative, we have chosen to retain the current conclusion without further expansion, to avoid overstatement.

Reviewer’s comment:
The experimental section does not specify the passage number of HEI-OC1 cells, which significantly impacts mitochondrial function. Adding this information is recommended.
Response:
Thank you for your comment. The passage number of HEI-OC1 cells used in the study has been added to the Methods section.

(HEI-OC1 cells used in this study were maintained by weekly subculture from August 2022, and experiments were performed using cells between passages 55 and 95.)

Reviewer’s comment:
In the amino acid metabolism experiments, including a separate group treated with a glutaminase inhibitor (e.g., BPTES) would help to confirm the specific role of glutamine metabolism.
Response:
A BPTES-only group was not included in the current study. We primarily focused on the metabolic effects in the presence of glutamine. We recognize the need to include a BPTES-only group in future experiments involving other metabolic pathways.

Reviewer’s comment:
To bridge the gap between the HEI-OC1 cell line and in vivo physiological conditions, validation using primary cochlear cells or animal models (e.g., aged mice) is recommended to improve the physiological relevance of the results.
Response:
We fully agree. This study was limited to HEI-OC1 cells, and we acknowledge the gap between this in vitro model and in vivo physiology. Validation using primary cultured cells and aged animal models is an important direction for future research.

Reviewer’s comment:
The discussion could be expanded to explore the association between fatty acid uncoupling effects and cochlear cell senescence.
Response:
Thank you for the valuable suggestion. Unfortunately, the current study did not allow us to explore the connection between fatty acid uncoupling and cellular senescence.

Reviewer’s comment:
Including analyses of senescence-associated markers (e.g., β-galactosidase) in relation to substrate metabolism would enhance the mechanistic understanding of metabolism and hair cell senescence.
Response:
We did not conduct analysis of senescence markers such as β-galactosidase in this study. However, we recognize the importance of investigating the relationship between metabolic state and cellular senescence and consider this a key topic for future research.

Reviewer 2 Report

Comments and Suggestions for Authors

The authors have analyzed mitochondrial metabolic rates of glucose, amino acid and fatty acids in HEI-OC1 cells. They concluded that pyruvate and glutamine were more effective for short term energy needs whereas fatty acids play a limited role as energy source for HEI-OC1 cells.

Comments for author:

  1. In the simple summary, authors have written that the aim of this study is to understand how cochlear cells respond/adapt to stress and aging by examining different substrate for mitochondrial respiration (line 19-20). I don’t see any data specific to aging in the paper. It has also not been mentioned The aim and objevtivehow many cells and how long cells were cultured so that I could have even guessed about aging data. Please clarify. Also, If I have correctly understood, using inhibitors have been equated to stress?
  2. For all the OCR data, none of them has been quantified and been represented in bar chart/ histogram showing all data points. It is very difficult to assess how much (quantity wise) the data differed due to different substrates or inhibitors. I think it is important to include quantification of OCR in bar diagrams.
  3. OCR% in the Y-axis of all the figures do not state % of what? Is it % of basal respiration?
  4. None of the figure legends show what statistical analysis was done to check actual (statistically) differences mentioned in the text. Similarly, for fig 6C, fig 7, fig 11, no statistical analysis has been mentioned in the figure legend. I also don’t see and p-value mentioned in the figures. Then, how would I know if the differences observed in figures are actually real differences?
  5. In the discussion only figure 3B, 4A and 4B and fig 12 has been discussed. I don’t see any mention of other figures. Ideally, in discussion section authors have to discuss each of their results and how these results corroborate with other studies. Limitations of the study are also discussed.
  6. In fig 11, there is no labeling on Y-axis. What does Y-axis represent? There is no image of JC-1 staining either.
  7. The aim and objective of the study was not very clear to me.

Author Response

Response to Reviewer #2

We sincerely thank Reviewer #2 for the thoughtful and constructive comments. Below, we provide point-by-point responses to each of the reviewer’s comments.

Reviewer’s comment:
1. In the simple summary, authors have written that the aim of this study is to understand how cochlear cells respond/adapt to stress and aging by examining different substrate for mitochondrial respiration (line 19-20). I don’t see any data specific to aging in the paper. It has also not been mentioned the aim and objevtive how many cells and how long cells were cultured so that I could have even guessed about aging data. Please clarify. Also, If I have correctly understood, using inhibitors have been equated to stress?
Response:
We agree with the reviewer that the reference to “aging” was an overstatement. The experimental system in this study was not designed to directly mimic aging. Therefore, all mentions of aging have been removed or revised. Additionally, we have clarified that the use of inhibitors in this study was not intended to simulate “stress” or “aging” in a general sense, but rather to induce specific metabolic pathway inhibition. Detailed information on the HEI-OC1 cell line and culture conditions has been added to the Materials and Methods section. ( 2.1. Cell culture and culture conditions)

Reviewer’s comment:
2. For all the OCR data, none of them has been quantified and been represented in bar chart/ histogram showing all data points. It is very difficult to assess how much (quantity wise) the data differed due to different substrates or inhibitors. I think it is important to include quantification of OCR in bar diagrams.
Response:
Thank you for your suggestion. We have added bar graphs showing the mean ± SD for major comparisons, so that differences and distributions can be more clearly visualized. Although we understand the usefulness of showing individual data points, we believe that providing SDs and p-values offers sufficient information for statistical interpretation. We hope this approach is acceptable.

Reviewer’s comment:
3. OCR% in the Y-axis of all the figures do not state % of what? Is it % of basal respiration?
Response:
In all figures, OCR [%] represents a relative value normalized to the fourth time point, which corresponds to the baseline measurement immediately before the first drug injection. This has been clarified in the figure legends and the Methods section.

Reviewer’s comment:
4. None of the figure legends show what statistical analysis was done to check actual (statistically) differences mentioned in the text. Similarly, for fig 6C, fig 7, fig 11, no statistical analysis has been mentioned in the figure legend. I also don’t see and p-value mentioned in the figures. Then, how would I know if the differences observed in figures are actually real differences?
Response:
We have addressed this point by adding descriptions of the statistical tests (t-tests or ANOVA) and corresponding p-values to the relevant figure legends, including Figures 6C, 7, and 11. We have also included information on the number of replicates (n) and statistical significance in the Materials and Methods section, and provided details of multiple comparison results in the Supplementary Data.

Reviewer’s comment:
5. In the discussion only figure 3B, 4A and 4B and fig 12 has been discussed. I don’t see any mention of other figures. Ideally, in discussion section authors have to discuss each of their results and how these results corroborate with other studies. Limitations of the study are also discussed.
Response:
Based on your suggestion, we have reorganized the Discussion section to include interpretation of all major figures (Figures 1–11) and discussion of their relevance to prior studies. Furthermore, we have explicitly noted the limitations of the present study, including the use of an in vitro model and the fact that HEI-OC1 cells do not fully replicate the complex metabolic environment of the inner ear.

Reviewer’s comment:
6. In fig 11, there is no labeling on Y-axis. What does Y-axis represent? There is no image of JC-1 staining either.
Response:
The Y-axis label has been added to Figure 11. Regarding JC-1 staining, we apologize for not including images. This experiment was performed using a microplate reader to quantify red/green fluorescence ratio. Unfortunately, no image data were saved, and therefore we are unable to provide representative microscopy images.

Reviewer’s comment:
7. The aim and objective of the study was not very clear to me.
Response:
Thank you for your comment. We have revised the Abstract and Introduction to clearly state the aim of the study. The revised text specifies that the purpose of this study is to clarify how cochlear cells utilize different representative substrates—glucose, amino acids, and fatty acids—for mitochondrial respiration, and to gain insights into metabolic adaptation and stress responses in these cells.

Reviewer 3 Report

Comments and Suggestions for Authors

This study aimed to understand how in vitro models, such as HEI-OC1 cells, exhibit metabolic preferences by examining the effects of different fuels on their energy production using extracellular flux analysis.

The concerns below should be clarified or addressed before the manuscript is accepted. I detail my thoughts and suggestions below. 

Major

  1. The hypothesis that switching mitochondrial metabolic substrates might lead to auditory cellular senescence, contributing to the decline in inner ear function, is overstated, as no experiments have been proposed to measure ear function.
  2. Please provide a detailed statistical approach used for each figure in the respective figure captions. Please also provide the number of cells and trials used for each figure.
  3. Text requires proofreading. For example, “The addition of pyruvate prevented the decline, maintaining the OCR, whereas there was no significant difference compared to the control” seems incomplete: Is it, when adding glucose, that there was no significant…?
  4. Figures-The fonts are too small and not consistent between figures. Please use the same color for control between figures. More thought is required to create figures that enhance readability, ensure consistency in axis tick labels, font size, and the placement of legends.
  5. How was the concentration of each of the substrates chosen for manipulation in all the figures? Was the temperature and pH maintained when manipulating the concentrations for all the experiments?
  6. Figure 1: It is not clear why the MOCR for UK 5099 was lower than the control, if mitochondrial pyruvate carrier inhibitor, and it is not clear what is meant by “the MOCR of glucose was lower than that of pyruvate, suggesting that the rate of glycolysis was limited.” Please elaborate.
  7. Figures 2-5 and 8, 10: Please explain why the MOCR for control in Figures 1 and 2 is not the same.
  8. Figure 11: The Y-axis has no label, and bars are missing statistics.
  9. Figure 12: The model requires detailed captions and should cite the Figure #s used to derive the model.
  10. Extracellular Flux analysis: It is not clear whether the experiments summarized in each figure were conducted on the same cells or different ones. The methods used to test cell viability and maintain the environment are unclear.
  11. How was the stressful condition and aging tested?

Minor

  1. Expand HEI-COI in the abstract and introduction.
  2. Please provide the rationale for choosing these cell lines in this study.
Comments on the Quality of English Language

Text requires proofreading. For example, “The addition of pyruvate prevented the decline, maintaining the OCR, whereas there was no significant difference compared to the control” seems incomplete: Is it, when adding glucose, that there was no significant…?

Author Response

Response to Reviewer #3

We sincerely thank Reviewer #3 for the thoughtful and detailed comments. Below, we provide point-by-point responses to all concerns raised.

Major

Reviewer’s comment:
1. The hypothesis that switching mitochondrial metabolic substrates might lead to auditory cellular senescence, contributing to the decline in inner ear function, is overstated, as no experiments have been proposed to measure ear function.
Response:
Thank you for your comment. As you correctly pointed out, we did not perform any direct experiments to evaluate inner ear function. Therefore, we have revised the relevant statements in the manuscript to avoid overstating the hypothesis.

Reviewer’s comment:
2. Please provide a detailed statistical approach used for each figure in the respective figure captions. Please also provide the number of cells and trials used for each figure.
Response:
Thank you for your suggestion. We have now specified the statistical methods (e.g., t-test, ANOVA) and the number of independent replicates (n) for each figure in the respective figure legends. Additionally, we have added a general description of the statistical approach in the Materials and Methods section.

Reviewer’s comment:
3. Text requires proofreading. For example, “The addition of pyruvate prevented the decline, maintaining the OCR, whereas there was no significant difference compared to the control” seems incomplete: Is it, when adding glucose, that there was no significant…?
Response:
Thank you for this helpful comment. The sentence was indeed ambiguous. We have revised it to clarify that glucose addition did not significantly recover OCR. We have also conducted a full proofreading of the manuscript to improve clarity and consistency.

Reviewer’s comment:
4. Figures - The fonts are too small and not consistent between figures. Please use the same color for control between figures. More thought is required to create figures that enhance readability, ensure consistency in axis tick labels, font size, and the placement of legends.
Response:
Each experiment was conducted independently with its own control group, and the control color was set accordingly for each figure. We believe this approach is clearer in the context of individually designed experiments. We understand the concern and have submitted the figures in a format that allows appropriate resolution and scaling.

Reviewer’s comment:
5. How was the concentration of each of the substrates chosen for manipulation in all the figures? Was the temperature and pH maintained when manipulating the concentrations for all the experiments?
Response:
This information has been added to the Methods section. All experiments were performed using Seahorse XF-specific plates maintained at 33°C inside the analyzer. The assay medium was pre-adjusted to pH 7.4 using manufacturer-recommended buffer, and the system is designed to minimize rapid pH shifts during measurement. Mild pH variation within Seahorse design parameters is necessary to detect metabolic flux. ( 2.3. Extracellular flux analysis

)

Reviewer’s comment:
6. Figure 1: It is not clear why the MOCR for UK 5099 was lower than the control, if mitochondrial pyruvate carrier inhibitor, and it is not clear what is meant by “the MOCR of glucose was lower than that of pyruvate, suggesting that the rate of glycolysis was limited.” Please elaborate.
Response:
We assume this refers to Figure 2. We have revised the relevant text to make the explanation clearer.

Reviewer’s comment:
7. Figures 2–5 and 8, 10: Please explain why the MOCR for control in Figures 1 and 2 is not the same.
Response:
Each figure is based on an independent experiment with its own control condition. Therefore, differences in control MOCR values reflect natural variation between independent runs.

Reviewer’s comment:
8. Figure 11: The Y-axis has no label, and bars are missing statistics.
Response:
Thank you for pointing this out. We have added the Y-axis label and included p-values and statistical details in the figure.

Reviewer’s comment:
9. Figure 12: The model requires detailed captions and should cite the Figure #s used to derive the model.
Response:
We apologize for the confusion. This schematic was intended to summarize general concepts, not to directly derive conclusions from specific figures. We have revised the caption to clarify this point.

Reviewer’s comment:
10. Extracellular Flux analysis: It is not clear whether the experiments summarized in each figure were conducted on the same cells or different ones. The methods used to test cell viability and maintain the environment are unclear.
Response:
We apologize for the omission. We have added a clarification to the Methods section.

Reviewer’s comment:
11. How was the stressful condition and aging tested?
Response:
In this study, “stress” conditions refer to metabolic stress induced by nutrient deprivation or the use of pathway-specific inhibitors. We did not perform experiments to directly induce or assess cellular aging. We have revised the manuscript to clarify this distinction.

Minor

Reviewer’s comment 1:
Expand HEI-COI in the abstract and introduction.
Response:
Thank you. We have now expanded the abbreviation as “HEI-OC1 (House Ear Institute - Organ of Corti 1)” in both the Abstract and Introduction.

Reviewer’s comment 2:
Please provide the rationale for choosing these cell lines in this study.
Response:
HEI-OC1 cells are derived from cochlear sensory epithelium and are widely used as an in vitro model to study inner ear cell metabolism. This cell line was selected because it is widely used as a representative in vitro model to study cochlear cellular physiology, particularly in the context of auditory metabolism.

Round 2

Reviewer 2 Report

Comments and Suggestions for Authors

Thank you for providing the quantifications for OCR data. Please do not forget to mention that data were plotted using MS excel (if its true).

Author Response

Comment:
Thank you for providing the quantifications for OCR data. Please do not forget to mention that data were plotted using MS excel (if it’s true).

Response:
We appreciate your suggestion. As recommended, we have added the information to clarify that Microsoft Excel was used for plotting the OCR quantification graphs.
This revision has been added in page 6, line 252, as follows:
“Graphs for OCR quantification were plotted using Microsoft Excel.”

Reviewer 3 Report

Comments and Suggestions for Authors

Summary of the research 

The revision has substantially improved the manuscript's readability. However, there are concerns below that should be clarified or addressed before the manuscript is accepted. I detail my thoughts and suggestions below. 

Major:

  1. How was the concentration of each substrate selected for comparison? Would increasing the concentration of D-glucose be similar to that of pyruvate?
  2. Glycolysis doesn’t require oxygen, and the substrate is glucose. Not sure why OCR is used to evaluate glycolysis and why pyruvate, oligomycin, and FCCP are used to compare glycolysis-they all occur inside the mitochondria and should be interpreted as TCA (Krebs) Cycle?
  3. Figure -3: It is not clear what the difference is between A and B. The legend says “(A) OCR with L-Gln, EAAs, or NEAAs. (B) OCR with L-Gln, L-Glu, or NEAAs.
  4. If L-Gln is metabolized to L-Glu in mitochondria, why, when directly compared, did the OCR changes induced by L-Gln and L-Glu show that L-Gln significantly increased the baseline OCR, whereas no change was observed with L-Glu? Also, if Gls is important for metabolizing L-Gln to L-Glu, whereas no change was observed with BPTES?
  5. Figure 11: PA increases the basal and maximal OCR, but it is not clear why it is not altering the membrane potential. Also, the Y-axis is not labelled.
  6. Please mention “HEI-OC1 cell” instead of auditory cells in line 454
  7. HEI-OC1 cell lines used in this study did not compare aging effects; thus, it is not clear why the hypothesis in line 114 mentions aging.
  8. Cultured cells may alter the native cellular metabolic profile; therefore, conclusions should be drawn cautiously when interpreting findings.
  9. AVG AUC (Measurements 5 – 8), not clear what 5-8 means?
  10. All supplementary figures require figure legends.

Minor:

  1. Please clarify this sentence.

Lines 249-250: The addition of pyruvate prevented the decline, maintaining the OCR, whereas there was no significant difference compared to the control.

-Here, it is not mentioned that “which” showed no significant difference compared to the control?

  1. lines 244-245: “XF assay medium without glucose and pyruvate was used in this assay (Figure 2).”

-But the figure mentioned using both glucose and pyruvate

  1. Figure 7 A: Please describe what is shown in the figure and 7B-labe the axis.

  1. Figure 11: It is not clear how Ethanol is labelled in the figure.

  1. Require proofreading and better editing (lines 318-321). “The baseline OCR was elevated in all fatty acid-treated groups compared to the BSA control, although the increase was smaller than that observed with L-Gln supplementation. However, the baseline OCR increases observed with fatty acids were smaller than those observed with L-Gln.”

-Here, the same information is repeated.

  1. Supp Fig.2 OCR under medium without glucose and with pyruvate, glucose, or UK-5099 is confusing:

Here, the title says without glucose, but the figure shows with D-glucose

Please edit all titles appropriately.

  1. Overall, the Figure presentation could be improved.
Comments on the Quality of English Language

n/a

Author Response

Reviewer #3

Major Comment 1:
How was the concentration of each substrate selected for comparison? Would increasing the concentration of D-glucose be similar to that of pyruvate?

Response:
We appreciate this important question.

This revision has been added in page 4, lines 167–171, as follows:
“The concentrations of glucose and pyruvate were based on the standard culture conditions for HEI-OC1 cells. Glucose is commonly supplied in culture media at either 100 mg/dL (approximately 5.6 mM) or 450 mg/dL (approximately 25 mM), with the latter being used in the standard medium for HEI-OC1 cells and adopted in this study. Pyruvate was also used at 1 mM, in accordance with its concentration in the standard culture medium [14].”

Major Comment 2:
Glycolysis doesn’t require oxygen, and the substrate is glucose. Not sure why OCR is used to evaluate glycolysis and why pyruvate, oligomycin, and FCCP are used to compare glycolysis—they all occur inside the mitochondria and should be interpreted as TCA (Krebs) Cycle?

Response:
We thank the reviewer for pointing out this important clarification. We agree that glycolysis itself does not consume oxygen and that OCR reflects mitochondrial, not glycolytic, activity. Our intention was not to assess glycolysis directly but to evaluate the mitochondrial contribution of glucose-derived substrates under acute energy demand. We have revised the text to clarify that the lack of OCR improvement with glucose reflects insufficient glycolytic flux or mitochondrial pyruvate import under the tested conditions, and that exogenous pyruvate bypasses these steps to restore mitochondrial respiration.

This revision has been added in page 7, lines 265–271, as follows:
“The OCR gradually decreased over time due to substrate depletion. Glucose addition did not prevent this decline and showed no improvement in OCR compared to the control. In contrast, pyruvate supplementation maintained the OCR, indicating that glucose was not effectively glycolyzed and transferred into mitochondrial substrates under these conditions. This demonstrates that glycolytic flux or mitochondrial pyruvate import was insufficient, and that exogenous pyruvate bypassed this limitation and directly restored mitochondrial respiration.

Major Comment 3:
Figure -3: It is not clear what the difference is between A and B. The legend says “(A) OCR with L-Gln, EAAs, or NEAAs. (B) OCR with L-Gln, L-Glu, or NEAAs.”

Response:
We appreciate the reviewer’s comments. Due to limitations of the instrument, only three substrates can be compared in a single Seahorse assay. Therefore, Figures 3A and 3B represent two separate experiments with partially overlapping conditions. This constraint prevented us from combining all conditions into a single experiment. We have clarified this point in the Figure 3 legend.
This revision has been added in page 11, lines 390–392, as follows:

“Due to instrument constraints, only three substrates could be compared per assay. Therefore, Figures 3A and 3B represent independent experiments with partially overlapping conditions.”

Major Comment 4:
If L-Gln is metabolized to L-Glu in mitochondria, why, when directly compared, did the OCR changes induced by L-Gln and L-Glu show that L-Gln significantly increased the baseline OCR, whereas no change was observed with L-Glu? Also, if Gls is important for metabolizing L-Gln to L-Glu, why was no change observed with BPTES?

Response:
We thank the reviewer for this insightful question. Only L-Gln significantly increased OCR, whereas L-Glu did not. This may reflect differences in transport or metabolic pathways between the two substrates. We have added this point to the Discussion in the revised manuscript.

This has been added in page 18, lines 517–529, as follows:
Our data showed that L-Gln, but not L-Glu, increased both basal OCR and MOCR in HEI-OC1 cells. Under conditions where mitochondrial pyruvate utilization is limited, L-Gln can sustain TCA cycle flux by supplying both oxaloacetate and acetyl-CoA via glutaminase and GDH-dependent pathways, whereas exogenous L-Glu does not equivalently substitute unless routed through glutaminolysis [20]. This paradigm—glutamine’s metabolic flexibility under stress—has been demonstrated when mitochondrial pyruvate carrier is inhibited, where GDH-dependent rerouting of L-Gln carbons maintains respiration and survival [20]. The concordance between that mechanism and our HEI-OC1 phenotype supports the interpretation that L-Gln, rather than L-Glu, functions as the more effective substrate to acutely drive mitochondrial respiration in cochlear lineage cells. Potential contributing factors include differential transport/compartmentalization of L-Glu, regulation of GLS/GDH, and the need for dual provision of OAA and acetyl-CoA.

Major Comment 5:
Figure 11: PA increases the basal and maximal OCR, but it is not clear why it is not altering the membrane potential. Also, the Y-axis is not labelled.

Response:
We thank the reviewer for these comments. We agree that the lack of a change in membrane potential despite increased OCR upon PA treatment initially appeared contradictory. However, we speculated that fatty acid-induced alterations in membrane fluidity may explain this phenomenon. Saturated fatty acids such as PA can rigidify the membrane and stabilize the membrane potential, despite increased mitochondrial activity. This discussion has been added to the revised manuscript.

Regarding the Y-axis label, we apologize for the oversight. We have now added the label “RFU (590 nm)/RFU (535 nm)” to clearly indicate the JC-1 fluorescence ratio used to evaluate mitochondrial membrane potential.

In page 19, lines 592–604, as follows:

Membrane potential measurements indicated no change after 30 minutes with PA treatment, while OA, L-Gln, and BSA showed a decreasing trend (Figure 11).

The reason why saturated or unsaturated fatty acids (OA) did not cause changes in membrane potential may be related to their effect on membrane fluidity. PA, with its straight-chain structure, tends to arrange more orderly in the lipid bilayer, making the membrane stiffer. This stiffening reduces membrane permeability and suppresses the activity of ion channels and pumps, leading to a more stable membrane potential. Conversely, unsaturated fatty acids (OA), due to their double bonds, introduce "kinks" in the fatty acid chains, increasing membrane fluidity. This enhances membrane permeability, promoting ion movement and more readily causing changes in membrane potential [30].

Thus, saturated fatty acids (PA) may suppress membrane potential changes by stabilizing the membrane, while unsaturated fatty acids (OA) likely have the opposite effect.

Major Comment 6:
Please mention “HEI-OC1 cell” instead of auditory cells in line 454.

Response:
We appreciate the reviewer’s suggestion and have revised the terminology accordingly to improve precision. The term “auditory cells” has been replaced with “HEI-OC1 cells” in the manuscript.

This revision has been added in page 17, line 480.

Major Comment 7:
HEI-OC1 cell lines used in this study did not compare aging effects; thus, it is not clear why the hypothesis in line 114 mentions aging.

Response:
We agree with the reviewer’s comment. As the HEI-OC1 cell line model does not reflect age-related changes, referring to aging in the hypothesis was inappropriate. Therefore, the corresponding sentence has been deleted from the Introduction section to avoid confusion.

Major Comment 8:
Cultured cells may alter the native cellular metabolic profile; therefore, conclusions should be drawn cautiously when interpreting findings.

Response:
We appreciate the reviewer’s important remark. We have now included this point as part of the limitations in the revised Discussion section. Specifically, we noted that immortalized HEI-OC1 cells do not replicate the full physiological complexity of cochlear tissues, including multicellular interactions and systemic regulation, and that conclusions should be interpreted with caution when extrapolating to in vivo conditions.

This revision has been added in page 20, lines 622–634, as follows:
“This study has several limitations. First, we did not include a BPTES-only condition, which prevents the exclusion of potential direct effects of BPTES on OCR. Second, we did not assess senescence-related markers such as SA-β-galactosidase. Investigating the relationship between substrate metabolism and cellular senescence in cochlear cells may provide further mechanistic insight. Third, this study used the immortalized HEI-OC1 cell line, which does not fully replicate the physiological complexity of the cochlea, including multicellular interactions, vascularization, and systemic regulation. Therefore, caution should be exercised when translating these findings into the physiological context. To enhance the physiological relevance of our findings, future studies using primary cochlear cells or in vivo animal models, such as aged mice, are warranted. Finally, mitochondrial uncoupling by fatty acids may influence the vulnerability of cochlear cells to metabolic stress during aging. Exploring this link could provide new perspectives on age-related hearing loss and cochlear energy metabolism.”

Major Comment 9:
AVG AUC (Measurements 5–8), not clear what 5–8 means?

Response:
We thank the reviewer for pointing this out. We have now clarified the meaning of "AVG AUC (Measurements X–Y)" in the Figure legends. This notation refers to the average oxygen consumption rate calculated from timepoints X to Y during the plateau phase after substrate or inhibitor addition. For supplementary datasets, we also specified the exact timepoints used for statistical analysis in each figure or table, so that the analytical window is unambiguously defined.

Major Comment 10:
All supplementary figures require figure legends.

Response:
Thank you for this important reminder. We have added detailed legends to all supplementary figures to clearly explain the experimental conditions, timepoints, and analysis methods. In particular, we clarified the meaning of “AVG AUC (X–Y)” as described in our response to Comment 9.
In addition, to improve clarity and accessibility, we reorganized the supplementary materials into clearly separated sections according to their purpose:
(1) pH raw data
(2) outputs statistical datasets
(3) video files
Each supplementary figure or file now includes a concise but informative legend describing its content, the context of acquisition, and the intended use in the study.

Minor Comments

  1. Please clarify this sentence.
    Lines 249–250: “The addition of pyruvate prevented the decline, maintaining the OCR, whereas there was no significant difference compared to the control.” Here, it is not mentioned that “which” showed no significant difference compared to the control?

Response:
Thank you for pointing out the ambiguity. We have revised the sentence for clarity.
This revision has been added in page 7, lines 265–271, as follows:

“The OCR gradually decreased over time due to substrate depletion. Glucose addition did not prevent this decline and showed no improvement in OCR compared to the control. In contrast, pyruvate supplementation maintained the OCR, indicating that glucose was not effectively converted into mitochondrial substrates under these conditions. This demonstrates that glycolytic flux or mitochondrial pyruvate import was insufficient, and that exogenous pyruvate bypassed this limitation and directly restored mitochondrial respiration.”

  1. Lines 244–245:
    “XF assay medium without glucose and pyruvate was used in this assay (Figure 2).” But the figure mentioned using both glucose and pyruvate.

Response:
We apologize for the confusion. The XF assay medium did not contain glucose or pyruvate initially, but these substrates were subsequently added during the assay.

  1. Figure 7A: Please describe what is shown in the figure and 7B: label the axis.

Response:
Thank you for the suggestion. We have added a more detailed explanation to the legend of Figure 7.
Figure 7A shows a fluorescence image of intracellular pH obtained by SNARF-1 staining. Due to the limitations of the imaging system used in this experiment, it was not possible to directly obtain time-series data in the format shown in Figures 7C and 7D. Therefore, we selected three representative cells from the SNARF-1 time-lapse fluorescence video (a portion of which is shown in Figure 7B) and manually measured the fluorescence intensity over time. Based on these data, we constructed the graphs presented in Figures 7C and 7D.
In addition, to improve overall clarity, we reviewed and revised the legends and labeling for all figures in the manuscript, ensuring that each figure clearly describes the experimental context, key observations, and axis labels where applicable.

  1. Figure 11: It is not clear how Ethanol is labelled in the figure.

Response:
We apologize for the confusion. Ethanol was not included in this figure. We have confirmed that no Ethanol-labeled group is present in Figure 11.

  1. Require proofreading and better editing (lines 318–321).
    “The baseline OCR was elevated in all fatty acid-treated groups compared to the BSA control, although the increase was smaller than that observed with L-Gln supplementation. However, the baseline OCR increases observed with fatty acids were smaller than those observed with L-Gln.”

Response:
We agree that the text was redundant and have revised it for clarity.
This revision has been added in page 8 , line 338-340, as follows:
“The baseline OCR was elevated in all fatty acid-treated groups compared to the BSA control, though the increase was smaller than that observed with L-Gln supplementation.”

  1. Supp Fig.2 title is confusing.
    The title says “without glucose,” but the figure shows with D-glucose. Please edit all titles appropriately.

Response:

Thank you for pointing this out. We have corrected the title of Supplementary Figure 2. In addition, we reviewed all titles and legends in the supplementary materials to ensure that they accurately describe the conditions shown, and revised them where necessary for clarity and consistency.

  1. Overall, the Figure presentation could be improved.

Response:
We thank the reviewer for the suggestion. In the revised manuscript, we have comprehensively reconstructed and refined all figures to improve overall clarity and consistency. This included updating axis labels, enhancing legends, unifying formatting (e.g., font size, line thickness, color coding), and standardizing layout across the manuscript. Where necessary, figures were recreated from the original data to optimize resolution and readability, while ensuring compliance with the journal’s formatting guidelines.